# TDHook: A Lightweight Framework for Interpretability

## Abstract

Interpretability of Deep Neural Networks (DNNs) is a growing field driven by the study of vision and language models. Yet, some use cases, like image captioning, or domains like Deep Reinforcement Learning (DRL), require complex modelling, with multiple inputs and outputs or use composable and separated networks. As a consequence, they rarely fit natively into the API of popular interpretability frameworks. We thus present TDHook, an open-source lightweight, generic interpretability framework based on `tensordict` and applicable to any `torch` model. It focuses on handling complex composed models which can be trained for Computer Vision (CV), Natural Language Processing (NLP), Reinforcement Learning or any other domain. This library features ready-to-use methods for attribution, probing and a flexible get-set API for interventions, and is aiming to bridge the gap between these method classes to make modern interpretability pipelines more accessible. TDHook is designed with minimal dependencies, requiring roughly half as much disk space as `transformer_lens`, and, in our controlled benchmark, achieves at least a $\times 2$ speed-up over `captum` when running integrated gradients for multi-target pipelines on both CPU and GPU. In addition, to value our work, we showcase concrete use cases of our library with composed interpretability pipelines in CV and NLP, as well as with complex models in DRL.

## 1 Introduction

The increasing complexity of Deep Neural Networks (DNNs) has raised significant challenges in understanding their decision-making processes and internal representations. Advancements in large language models (Radford & Narasimhan, 2018), computer vision systems (Simonyan & Zisserman, 2014), and other deep learning architectures have further exacerbated the interpretability challenge by mixing modalities and domains, making it increasingly difficult for researchers and practitioners to debug, let alone trust, these systems. While the field of explainable AI has grown substantially in recent years, existing efforts in model interpretability predominantly focus on specific domains and architectures. Methods for attribution (Simonyan et al., 2013; Sundararajan et al., 2017), latent manipulation (Zou et al., 2023; Cunningham et al., 2023), and weights-based analysis (Dunefsky et al., 2024) have been well-developed, and frameworks like `captum` (Kokhlikyan et al., 2020), `zennit` (Anders et al., 2021), and `pyvene` (Wu et al., 2024) provide specialised tools for these techniques.

However, these solutions can struggle with complex interpretability pipelines, as modern works like concept relevance propagation (Achtibat et al., 2022) or attribution patching (Nanda, 2022) leverage disparate methods. They only offer limited compatibility with composed models with multiple inputs or outputs that often require heterogeneous data streams, such as common multi-modal domains (Radford et al., 2021; Alayrac et al., 2022), DRL (Liang et al., 2017) or other less general domains like finance (Zhou et al., 2025). Furthermore, these existing frameworks often lack support for `tensordict` structures, are limited to specific model architectures, and are not necessarily lightweight or efficient. These gaps leave practitioners without generic and efficient tools for understanding complex deep learning systems across various domains.

We thus propose TDHook, a lightweight, generic interpretability framework designed to address these limitations. TDHook implements post-hoc methods that can generate explanations directly from DNNs without constraining their design or needing to extract an interpretable model. The framework is built on two key design principles: (1) compatibility with any `torch` model through a flexible hooking mechanism,

and (2) native support for `tensordict` structures, which naturally model interpretability by-products like attributions, activations, gradients, and weights. We illustrate the high-level architecture of the framework in Figure 1, underlining the modularity and broad applicability of the framework.

TDHook makes it easier to compose complex interpretability pipelines across diverse domains, as demonstrated by our use cases study. Furthermore, it is designed with minimal dependencies, requiring roughly half as much disk space as `transformer_lens`, and, in our controlled benchmark, achieves up to a $\times 2$ speed-up over `captum` when running integrated gradients for multi-target pipelines on both CPU and GPU. We list our contributions as follows:

- Introduction of a new interpretability framework, TDHook[1], based on `torch` (Ansel et al., 2024) and `tensordict` (Bou et al., 2023).

- A set of more than 25 ready-to-use methods for attribution, latent manipulation and weights manipulation, unified through a common interface and only requiring a few lines of code to set up.

- A comparative analysis with existing frameworks through stress testing.

- Concrete use cases of TDHook applied to language, vision, and DRL models, illustrating multi-step interpretability pipelines.

This paper is structured as follows: we present an initial background about interpretability methods and frameworks in Section 2. Then, we describe the TDHook framework with its design principles in Section 3, before presenting a comparison study and showcasing concrete use cases in Section 5. Finally, we discuss the limitations and provide guidance on when to use TDHook in Section 6.

## 2 Background

### 2.1 Interpreting Deep Neural Networks

As we are interested in methods that can be directly applied to a Deep Neural Network (DNN) model, we take a post-hoc approach. In Computer Vision (CV) and Natural Language Processing (NLP), many approaches were developed in order to study pre-trained models (Simonyan & Zisserman, 2014; He et al., 2015; Radford & Narasimhan, 2018). In order to improve code modularity in the TDHook library, we use a categorisation of these methods based on their data stream and produced artifacts.

**Attribution methods**   These are typical methods used in CV to understand convolutional networks by visualising important pixels, i.e. saliency maps, (Simonyan et al., 2013). The key idea is to compute the sensitivity of the output or any intermediate representation with respect to the input or other intermediate representation. The first developed class of methods were gradient-based (Simonyan et al., 2013), with derived methods like integrated gradients (Sundararajan et al., 2017) or grad-CAM (Selvaraju et al., 2016), that are mathematically grounded with sensitivity. Certain methods compute importance maps by occluding part of the input (Zeiler & Fergus, 2013) or, more generally, perturbing the input (Covert et al., 2020). Other backpropagation methods like LRP locally define relevance rules, using deep-Taylor expansions, which enable global attribution (Montavon et al., 2015; Bach et al., 2015). Recent works in NLP focus on the Transformer architecture and its attention mechanism (Vaswani et al., 2017), providing token-level insights (Wiegreffe & Pinter, 2019; Achtibat et al., 2024). Finally, some methods go beyond single prediction explanation and aim to provide more conceptual explanations, like the computation of pre-images from activation maximisation (Mahendran & Vedaldi, 2015), or the finding of maximally activating samples in a dataset (Chen et al., 2020).

**Latent manipulation**   This class of methods techniques extends the interpretability of concepts and features by exploring the internal representations learned by models. Latent concepts were introduced in CV with (Kim et al., 2018), where concepts are defined as latent space directions, typically computed with linear probes (Alain & Bengio, 2018; Dreyer et al., 2023b). This field extends the notion of prototypes like

---

[1]The library is provided as supplementary material

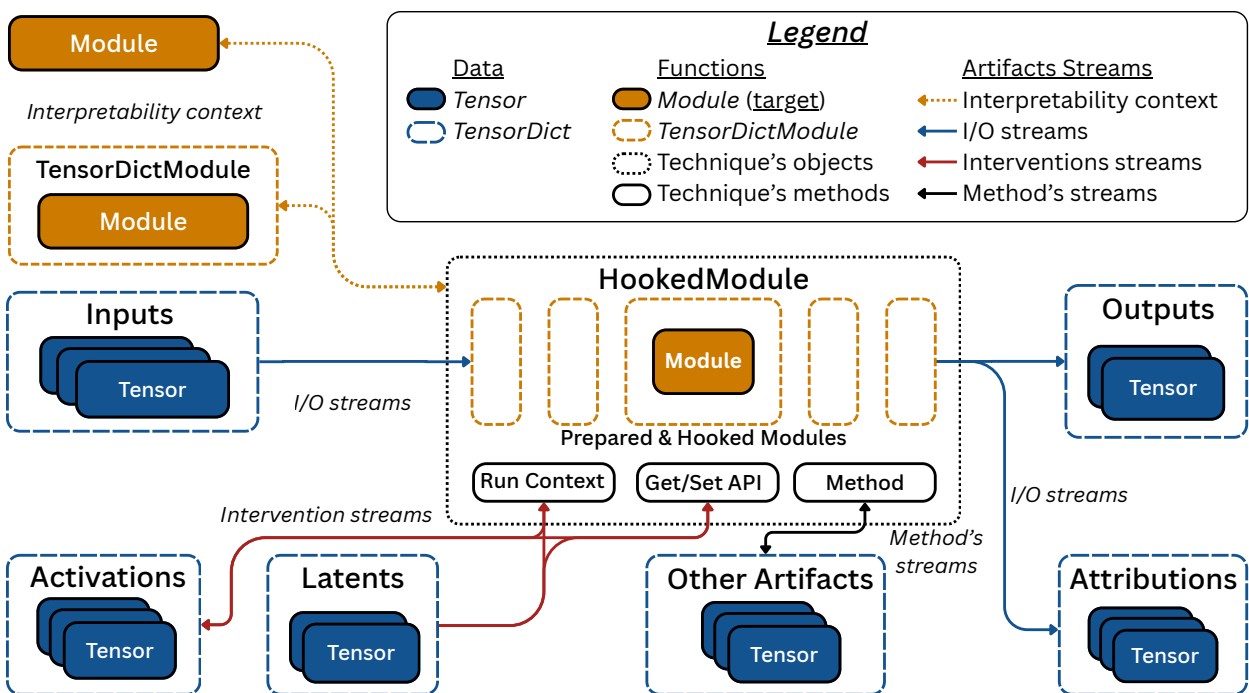

Figure 1: Schematic view of the TDHook framework architecture. The target to interpret is a `torch` module or a `TensorDictModule` wrapped in a `HookedModel`. This object exposes different APIs, like the get-set API described in Section 3.3, while its forward can be modified to fit the needs of different interpretability techniques, see an example in Section 3.2. The artifacts produced and consumed by the models and methods are materialised as `TensorDict` objects. More details about the design principles are given in Section 3.1, while interactions between the main components of the framework are given in Appendix D.

perturbed images (Ribeiro et al., 2018), cropped images (Dreyer et al., 2023a) or pre-images (Mahendran & Vedaldi, 2015). These methods were later expanded by the field of representation engineering (Zou et al., 2023), where such latent features enable locating, editing, erasing or decoding models' knowledge (Meng et al., 2022; Belrose et al., 2023; Ghandeharioun et al., 2024). Other works enable causally modifying or analysing the model outputs (Rimsky et al., 2023; Kram'ar et al., 2024), or use sparse autoencoders to elicit interpretable features in language models (Cunningham et al., 2023).

**Weights-based methods** These methods focus on interventions and analysis of model parameters themselves, rather than activations or gradients (Saphra & Wiegreffe, 2024). These methods provide a more granular understanding of model internals by examining pathways and dependencies between models' components, usually neurons or attention heads, and aim to remove the data dependency from explanations in order to provide out-of-distribution explanations. Viewing models' components as circuits was first proposed for CNNs (Olah et al., 2020) before being formalised for Transformers (Elhage et al., 2021). These circuits revealed peculiar models' components that learned precise mechanisms like induction (Olsson et al., 2022). Using specific datasets, relevant circuits can be automatically discovered (Conmy et al., 2023), and can also involve larger models' components at the layer scale (Dunefsky et al., 2024). In this framework, operations on the weights become natural, like editing using task vectors (Ilharco et al., 2022; Hendel et al., 2023). As a consequence, it is also possible to use explanations, from attribution or circuit discovery, to selectively prune models (Yeom et al., 2019; Pochinkov & Schoots, 2024).

## 2.2 Interpretability Frameworks

**Frameworks landscape** A range of open-source libraries has emerged to operationalise some of the previously described techniques, each with its own focus and design philosophy. `captum` (Kokhlikyan et al.,

Table 1: Comparison of interpretability frameworks on selected features. **Any Torch**: compatibility with arbitrary PyTorch models; **TensorDict**: support for `TensorDict` and `TensorDictModule`; **Get-Set**: availability of a generic intervention API; **Composable**: ability to compose different methods together; **Methods**: number of ready-to-use interpretability methods; **Coverage %**: percentage of the codebase covered by automated tests.

| Library | Any Torch | TensorDict | Get-Set | Composable | Methods | Coverage % |
|---|---|---|---|---|---|---|
| captum | ✓ | ✗ | ✗ | ∼ | 35+ | 93% |
| inseq | ✗ | ✗ | ✗ | ∼ | 15+ | - |
| nnsight | ✓ | ✓ | ✓ | ✗ | 0 | - |
| pyvene | ✓ | ✗ | ✓ | ✓ | 0 | - |
| transformer_lens | ✗ | ✗ | ✗ | ✗ | 0 | 75% |
| zennit | ✓ | ✗ | ✗ | ✓ | 15+ | 99% |
| tdhook | ✓ | ✓ | ✓ | ✓ | 25+ | 95% |

2020) is the most widely-used `torch` interpretability library, in terms of downloads and GitHub stars, and concentrates on attribution, providing more than thirty gradient and perturbation-based explainers. `zennit` (Anders et al., 2021), the de facto library for Layer-wise Relevance Propagation methods, offers a unified interface for relevance-based explanations. In the causal-analysis corner, `pyvene` (Wu et al., 2024) provides configuration-based composable interventions while `nnsight` (Fiotto-Kaufman et al., 2024) exposes a general intervention API, based on a getters and setters (get-set API), for altering models' computations at inference time. `inseq` (Sarti et al., 2023) specialises in sequence-generation models attribution, most notably by porting the methods from `captum`. Finally, `transformer_lens` (Nanda & Bloom, 2022) targets Transformer models and eases their study by providing a unified access to different model architectures.

**Frameworks comparison**  In Table 1, we compare these frameworks on features selected to highlight the `tdhook` library's strengths. These features are often missing in the other frameworks and having them all in one place was one of the motivations behind `tdhook`. While `captum` excels in attribution methods, it lacks support for `TensorDict` and provides no intervention capabilities, limiting its applicability to modern PyTorch workflows and causal analysis. `nnsight` offers a powerful get-set API but provides no built-in interpretability methods, requiring users to implement techniques from scratch. `pyvene` supports interventions but lacks `TensorDict` integration. `transformer_lens` is specialised for Transformer architectures only, excluding other model types. `zennit` focuses solely on LRP methods, while `inseq` is restricted to sequence generation models.

`tdhook` combines all these capabilities: it supports arbitrary PyTorch models with `TensorDict` integration, provides a comprehensive get-set intervention API, offers composable interpretability methods, and includes more than 25 ready-to-use techniques with extensive test coverage.

## 3  TDHook

We now describe the TDHook framework, which is summarised in Figure 1.

### 3.1  Design Principles

Our objective to create **composable** interpretability methods in order to handle complex pipelines leads us to consider a **TensorDict-powered** approach, which provides a strong standard for data and model manipulation. We also aim to provide **ready-to-use** interpretability methods, making them easy to use even by non-expert users, while keeping the flexibility of a **generic** framework. Finally, we want our framework to be **lightweight** for maximal portability.

**Composable**  One of the first objectives of TDHook is to be able to implement complex interpretability pipelines, like concept relevance propagation (Achtibat et al., 2022) or attribution patching (Nanda, 2022). While these methods can be decomposed into simpler methods, providing a simple interface to run them

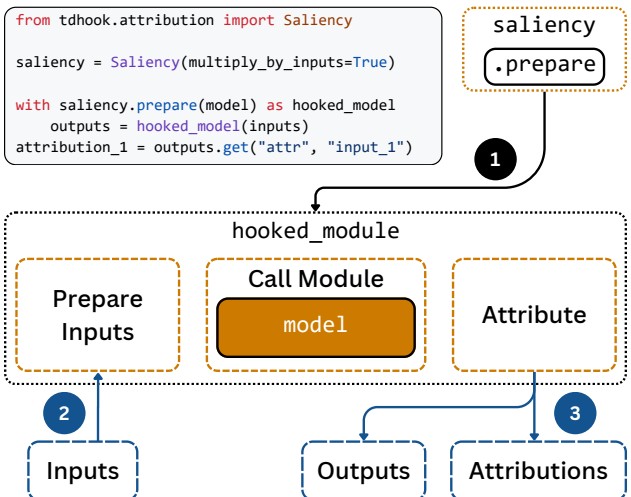

Figure 2: Schematic view of the code execution for the Saliency method in TDHook. (1) First, we define a context factory that will prepare a `Module` or `TensorDictModule` into a `HookedModel` instance. (2) Then, we run this modified model inside the context with `TensorDict` inputs. (3) Finally, we can retrieve the attributions from the output `TensorDict`.

makes them more accessible. For that, we abstract these methods using a unified approach to how models and data are manipulated, allowing for easy composition. This is made possible by the `tensordict` library.

**TensorDict-powered**   The `tensordict` library is designed to manage groups of tensors and multi-input/output models (Bou et al., 2023). It provides two primitives: `TensorDict` for carrying data and `TensorDictModule` for operations on `TensorDict` like model predictions. We focus our efforts around the following postulate: interpretability methods can be implemented in terms of `TensorDictModule` operating with `TensorDict` artifacts. Indeed, it is clear that interpretability by-products like activations, gradients, weights or attributions are collections of tensors, i.e. `TensorDict`, with different key types (i.e. module names, parameter names, input names, etc.). And therefore interpretability methods are just `TensorDictModule`, e.g. attribution methods take in input a `TensorDict` (inputs) and return a `TensorDict` (attributions).

**Ready-to-use**   Similarly to `captum` (Kokhlikyan et al., 2020), we focus our efforts on making interpretability methods easy to use, and provide a wide range of methods, ready to be used even by non-expert users. This strength is especially important in order to democratise the interpretability of DDNs. In that respect, TDHook is not only a generic interpretability framework but also a pragmatic and easy way to integrate interpretability methods. An example using a ready-to-use method is given in Figure 2 and a list of the currently implemented methods is given in Appendix D.

**Generic**   TDHook is a generic framework, compatible with any Pytorch model, like `captum` or `nnsight`. It enables users to directly plug their model into the framework and to get started rapidly. While we mention our design around the `tensordict` library, we remain compatible with `Module` from `torch` as they can be simply wrapped with the `TensorDictModule` class. Furthermore, in order to enable broader and more customisable methods, we replicate a flexible intervention API similar to `nnsight`, based on getters and setters (get-set API), of which we provide an example usage in Figure 3.

**Lightweight**   Some frameworks like `transformer_lens`, `pyvene`, or `inseq` find their strength in being tailored to work with Transformers, but it comes at a dependency or incompatibility cost. Yet most of these frameworks rely on `torch` hooks at their core, which provide the fundamental mechanism for interpretability. In this respect, TDHook is built around `torch` hooks in order to be a lightweight interpretability framework with minimal dependencies. By only admitting two dependencies, `torch` (already present to run the model)

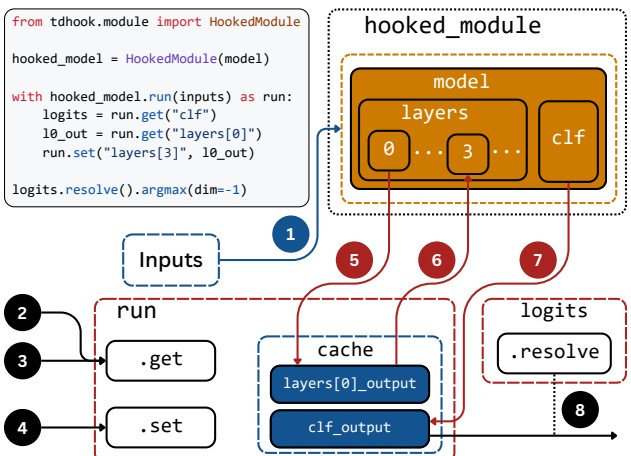

Figure 3: Schematic view of the code execution for an intervention in TDHook. (1) After explicitly defining a `HookedModel` instance, we enter a run context with the given inputs. (2-4) We query the run instance to define our intervention scheme which registers the required hooks on the model. This get-set API returns cache proxies that can be retrieved upon model execution. (5-7) At the context exit, we execute the model, which triggers the registered hooks and populates the cache. (8) Finally, we can retrieve any intermediate state by resolving the proxies previously defined.

and `tensordict`, we drastically limit the potential dependency conflicts and the bundle size of the library. We provide a benchmark of the bundle size of different interpretability libraries in Section 4.

### 3.2 Ready-to-use Methods

Ready-to-use methods are a powerful adjunct to any library as they enable non-experts to apply interpretability methods. These methods encapsulate common interpretability workflows into simple function calls, abstracting away the complexity of manually setting up hooks, interventions, and data processing pipelines. By providing pre-configured implementations for methods like saliency mapping, latent manipulation, and weights manipulation, users can focus on their research questions rather than implementation details. The underlying complexity of hooking mechanisms, tensor manipulation, and result formatting is handled automatically, making interpretability accessible to researchers with varying levels of technical expertise. This is especially relevant for cross-disciplinary research where DNNs, used as black-box tools, would benefit from being explained. Figure 2 illustrates the execution flow for a saliency method, showing how the context factory prepares the model and how attributions are retrieved. For comprehensive implementation details and code examples, we refer the reader to Appendix D and the supplementary materials.

### 3.3 General Get-Set API

As already proposed by (Fiotto-Kaufman et al., 2024), the get-set API is a versatile way to interact with a model. This pattern separates the definition of interventions (with getters and setters) from their execution and result retrieval. We adopt a more explicit approach using `HookedModel` methods that provide fine-grained control over model interventions and state retrieval. The get-set API allows researchers to first define their intervention scheme, specifying which layers to hook, what modifications to apply, and which intermediate states to capture, before executing the model. This separation enables complex intervention strategies to be built incrementally and ensures that all interventions are applied consistently during a single forward pass and a potential backward pass. The API returns cache proxies that act as placeholders for future values, which are then populated during execution and can be resolved afterwards to retrieve the actual intermediate representations. The model can also be executed in a run context offering the same API, triggering the registered hooks and populating the cache with the intermediate representations. This context automatically handles hooks deregistration, temporary cache configuration and offers other capabilities like stopping the

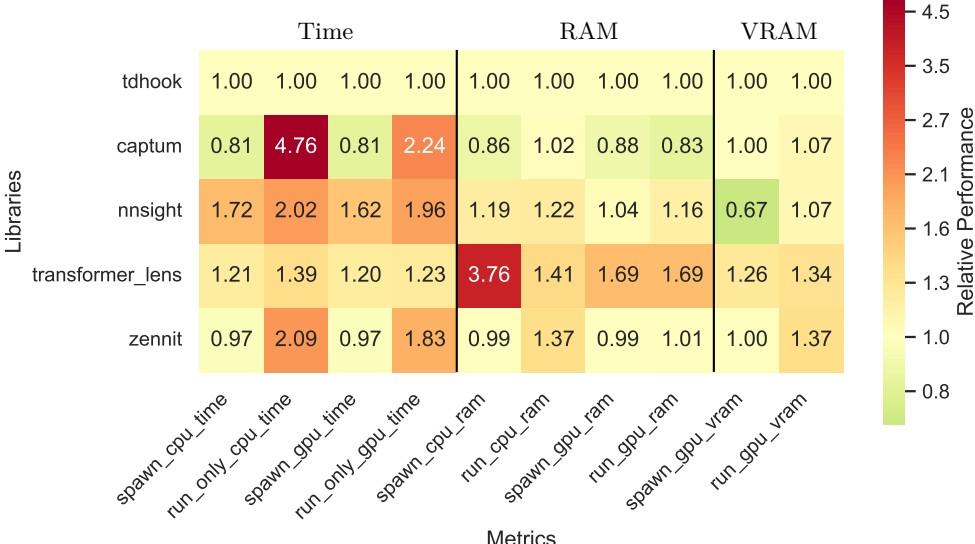

Figure 4: Relative performance of interpretability frameworks on their "Get Started" task compared against tdhook implementation (lower is better). The colours follow a logarithmic scale centred on the tdhook relative performance. More technical details and results are given in Appendix A, and the individual task results are provided in supplementary material.

computation at any layer. Figure 3 demonstrates the intervention workflow, showing how hooks are registered during the setup phase and how intermediate states are retrieved after execution. Implementation details can be found in Appendix D and in the supplementary materials.

## 4 Benchmarking

In order to compare our library beyond the accuracy of the implemented methods, we introduce two benchmarks to position our library in the framework landscape.

**Performance benchmark**   In order to fairly compare the different libraries, we use the tasks which are showcased as a "Get Started" task in their respective documentation, and we reproduce them in TDHook, checking the consistency of the results[2]. For each task, we run multiple experiments with different seeds, model sizes, batch sizes and other parameters, consisting of more than a hundred variations. Our experiments include models ranging from small MLPs to transformers (using the GPT-2 family), with up to 1B parameters. The experiments were conducted with batch sizes up to 100k for MLPs, in order to explore the limits of common use cases like activation caching. For reproducibility purposes, we give more details in Appendix A and provide the complete code in the supplementary material. We also depict the exact hardware configuration used for the benchmark in Appendix A. In Figure 4, we report the aggregated results of our benchmark. We measure the relative performance of each library against TDHook across different metrics (time, RAM, VRAM) and different setups (CPU only or CPU+GPU). Each task run is performed in isolation and always starts with a "spawn" (model setup or other preparations without computation) for which the metrics are also reported. This kind of benchmark is useful to stress-test the different libraries and to highlight the different trade-offs between them in terms of resource consumption. For example, while Captum is optimised in terms of memory usage, it is slower due to its gradient computation strategy[3], it does not scale well with the batch size. TransformerLens has a setup overhead in terms of memory usage in order to convert the model to a unified format; it does not scale well with the model size.

---

[2]Caching in `transformer_lens` is not reproduced exactly due to the model modifications needed.

[3]Captum only computes the gradient with respect to scalar values instead of vector values.

**Bundle size benchmark**    The second benchmark compares the bundle size of different interpretability frameworks. We considered two different metrics, which are synonyms of light-weight disk usage: the memory space and the number of inodes. While the disk space is commonly used to partition the disk among users, the number of inodes can also be limited in certain scenarios, like HPC clusters. The results in Figure 5 differentiate libraries that focus on being model agnostic versus libraries that focus on models from the Transformers library (Wolf et al., 2020). These measurements are conducted in the independent virtual environments obtained by installing each framework individually. We managed those dependencies by using `uv` (Astral Software Inc, 2025); more details about the benchmarks are given in Appendix A.

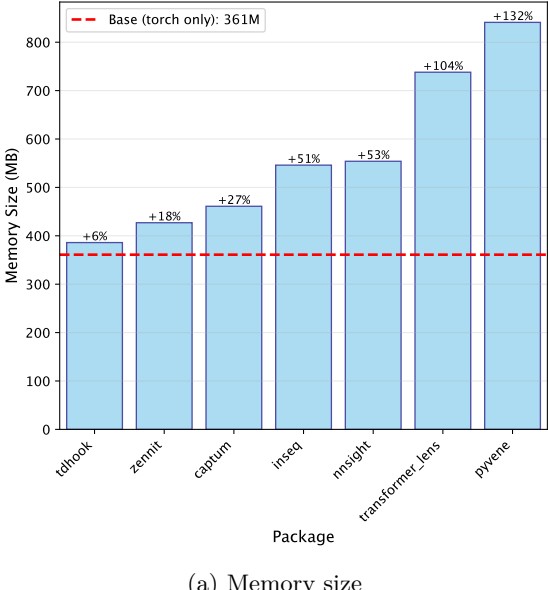

(a) Memory size

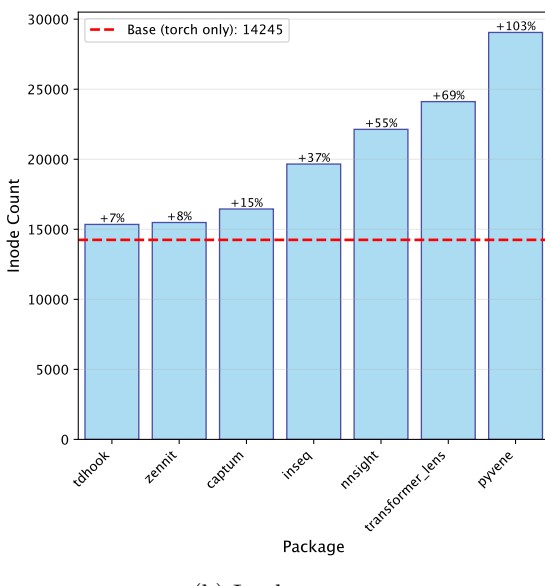

(b) Inode count

Figure 5: Bundle size comparison across different interpretability frameworks by measuring the memory size (a) and the inode count (b). For each measurement, we compare with only installing `torch`; this baseline space could also be reduced by choosing a lighter version of `torch`, e.g. `torch-cpu`.

## 5    Use Cases

We now showcase examples demonstrating specific features that motivated the design of TDHook, and extend results from previous works. These examples are spread across natural language processing, computer vision and reinforcement learning.

### 5.1    Complex Pipelines

Modern interpretability pipelines have evolved into complex multi-step procedures, mixing different methods in order to derive more comprehensive explanations. We here focus on concept attribution and attribution patching, which are described in the following paragraphs.

**Concept attribution**    Contrary to traditional attribution methods that focus on explaining the output of the model, e.g. the class predicted, with respect to the input, concept attribution methods focus on explaining the output with respect to specific concepts, illustrated in Figure 6. Most of these methods rely on conditionally propagating the attribution with respect to specific channels, like conductance for integrated gradients (Dhamdhere et al., 2018) or concept relevance propagation for LRP (Achtibat et al., 2022). These channel concepts are usually discovered or visualised using activation maximisation (Chen et al., 2020) or relevance maximisation (Achtibat et al., 2022). But since concept activation vectors (CAVs) generalise these methods, we introduce a propagation method based on the TCAV scores (Kim et al., 2018). Different methods

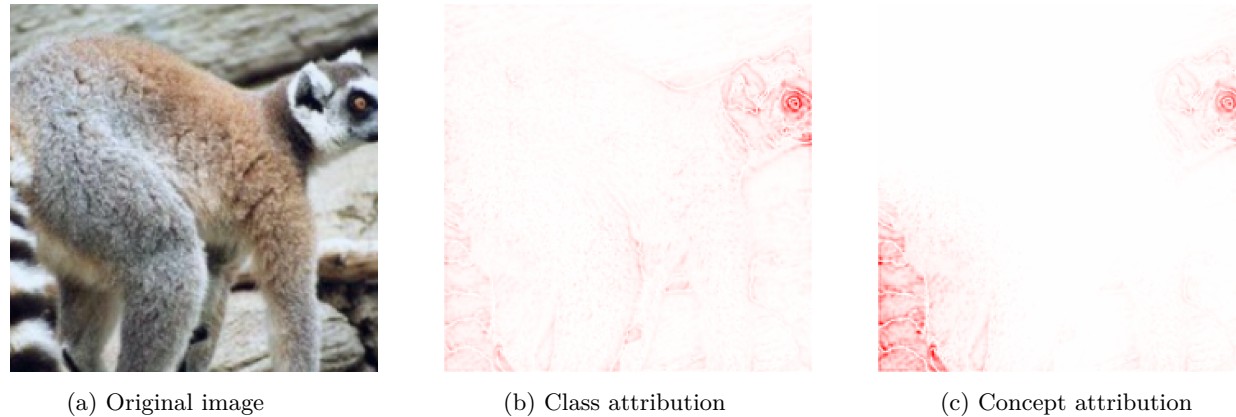

(a) Original image  (b) Class attribution  (c) Concept attribution

Figure 6: Concept attribution with the LRP method. (a) The original image being explained. (b) The class attribution heatmap. (c) Attribution of the channel associated with the concept "striped" selected by activation maximisation.

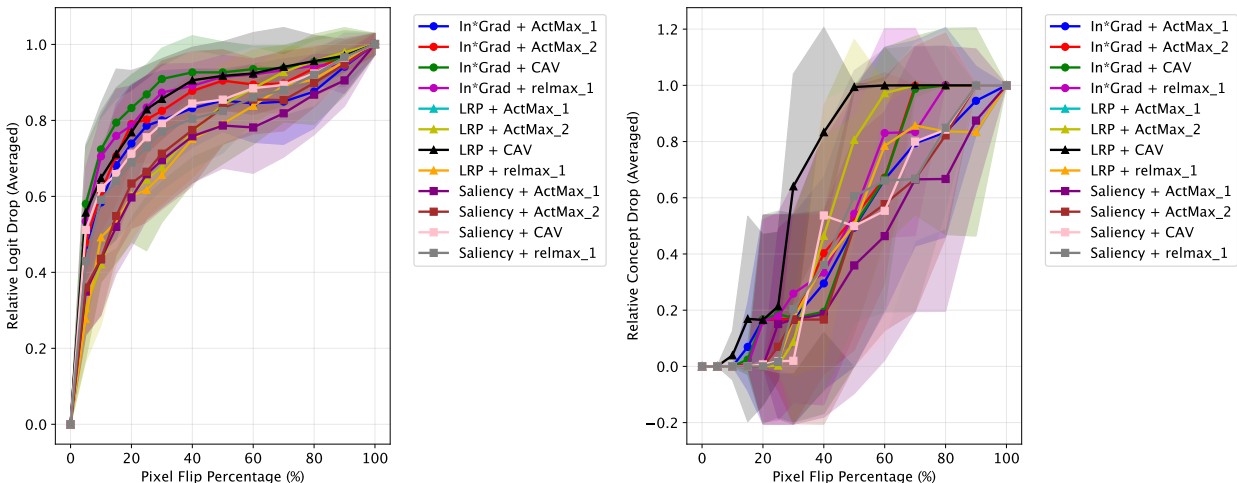

Figure 7: Concept attribution evaluation for the concept "striped". We report the change in the class logit and the concept probability when the input is randomised according to the relevance scores. The ablated pixels are replaced by the mean of the input image.

for producing CAVs were already compared in (Dreyer et al., 2023b), but not in the context of concept attribution.

Concept attribution is complex, as it is a composed method, and it is important to be able to compare across a variety of possible combinations of methods. As all methods are implemented in TDHook, it is possible to easily compare them by simply switching the method used, enabling users to chose their prefered notion of concept or attribution. For our experiments, we use the VGG16 model (Simonyan & Zisserman, 2014), and extracted concepts like "striped" using the texture dataset from (Cimpoi et al., 2013). In order to evaluate the methods, we randomise the input according to the relevance scores, i.e. pixel flipping (Bach et al., 2015), and record the change in the class logit and the concept probability. The results, reported in Figure 7, show that using the TCAV scores gives attribution that is more faithful to the concept. TCAV marginally outperforms the other methods according to pixel-flipping evaluation, as shown in Figure 7. Yet, qualitative analysis suggests that the obtained attribution is not localised, e.g., showing relevance spread beyond the lemur's tail. Therefore other metrics should be used to evaluate the quality of the attribution, like localisation or robustness metrics (Hedström et al., 2022).

**Attribution patching**   Activation patching, also known as causal mediation analysis (Vig et al., 2020), is a method to understand the contribution of different components of the model to the output. It provides a powerful causal analysis but is costly to compute, as it requires as many forward passes as there are components to patch. Attribution patching (Nanda, 2022) was proposed as an alternative using a first-order approximation, only requiring two forward passes and one backward pass. Recent work has replaced the gradient by a relevance score obtained using LRP, i.e. RelP (Jafari et al., 2025). We explore the generalisability of this idea by replacing gradients by latent relevances. And show that the results of the RelP method are sensitive to the LRP rules used, notably the results using the AH-rule, introduced in the paper Jafari2025RelPFA but not used for the experiments, are presented in Appendix B. The results in Figure 8 show that only methods closely related to the gradient are relevant for attribution patching.

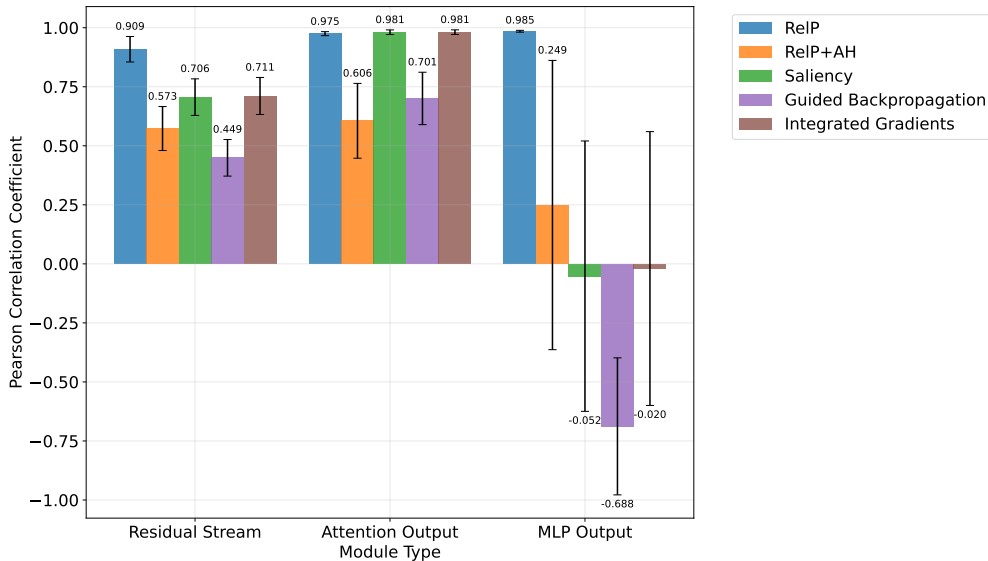

Figure 8: Pearson correlation coefficients between different attribution patching methods and the activation patching method. The values are averaged for different models (GPT-2 family), and the error bars represent the standard deviation. This metric does not account for amplitude differences between the methods.

## 5.2   Complex Models

In addition to the pipelines, models can also expose a complex structure, with multiple inputs and outputs. We here focus on multi-output models, which are common in DRL.

**Analyse multi-output models**   It is common for models in RL to have multiple inputs and outputs. We here take the example of a chess model which is trained with AlphaZero (Silver et al., 2018). In particular, we choose a network from McIlroy-Young et al. (2020), which outputs the policy over the legal moves and the win-draw-lose probabilities for the current position. More details about the models used are given in Appendix B. In Figure 9 we show saliency maps for the policy and the value heads obtained using gradient attribution (Simonyan et al., 2013). The position chosen, taken from team (2025), is a mate in 2 for the black player (`e3e1`, `b1a2`, `e1a1`). Attributing the best move gives a positive saliency to the rook and the bishop, while the adversary king has a negative saliency. This reflects the move sequence leading to the mate: the rook checks the king, the king flees to the side, and the rook mates while being defended by the bishop. For the win probability attribution, the attacking pieces have a high positive saliency, while the black king and the pawn have a negative saliency, as they are weak points. The adversary queen also has a high positive saliency, as it can be captured by the knight. While the predictive power of the saliency is limited, it can still provide basic insights about the model.

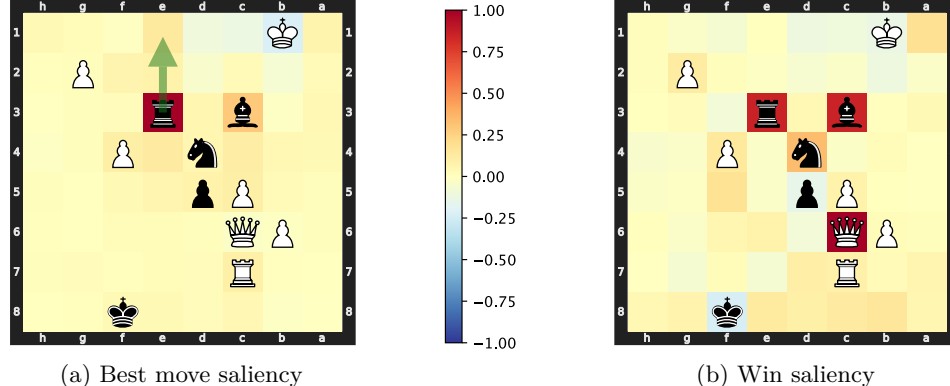

(a) Best move saliency             (b) Win saliency

Figure 9: A same board position analysed through the prism of the policy head and the value head using gradients. The attributions are scaled by the absolute maximum value such that the colour meaning is shared for (a) and (b). For policy attribution (a), the output explained is the logit of the best legal move, which is plotted as a green arrow, and for the value attribution (b), the output explained is win probability.

**Analyse composed models** While the previous example showcased a shared backbone for the policy and the value, it is not always the case. Especially in decentralised multi-agent settings, the value is often independent from the policy, and each agent can have its own policy and value. Therefore, it means that you have multiple networks to explain. Yet, in TorchRL, these models are grouped in a single `TensorDictModule`, the loss, which can be directly wrapped in TDHook. For our illustration, we train an agent with PPO (Schulman et al., 2017) on the inverted double pendulum environment (Todorov et al., 2012). More details about the agent studied are given in Appendix B. In this case, we decided to explore how the predicted action is represented throughout the internal states of this agent using linear probing (Alain & Bengio, 2018). Figure 10 shows the action probing in the agent, performed both on the policy and the value by wrapping the loss module. For each layer (pre- and post-activation), a linear regression probe is trained to predict the model's action. We can see that the policy head contains more information about the action than the value head, as expected, but the value head still overperforms the baseline trained from the raw observations. For the policy, the last layers contain less information about the predicted action because it is highly compressed, only predicting the mean and standard deviation of a Tanh-Normal distribution. For the value, the information about the action is not trackable in the last layers, and the probe overfitted on the training data, as it can be expected when the signal weakens.

## 6 Discussion

We now discuss the positioning of TDHook in the broad framework landscape with regard to its relevance to specific use cases, and propose avenues for future work.

### 6.1 When to Use TDHook

TDHook is most beneficial in scenarios where researchers need to analyse models that produce multiple outputs or are composed of several sub-modules, a situation frequently encountered in reinforcement-learning pipelines. Because the library natively embraces `TensorDict` and `TensorDictModule`, it can ingest the heterogeneous tensors emitted by these models without any additional data-wrangling layer. Its generic get-set API also makes TDHook a compelling choice for rapid prototyping: a new attribution or intervention can often be implemented in a handful of lines while retaining compatibility with the core abstractions. It is also relevant to build composed interpretability pipelines based on multiple ready-to-use methods which can be chained together. Finally, our benchmarks in Section 4 demonstrate that TDHook maintains a small installation footprint and competitive runtime memory usage; this makes it a sensible default on resource-constrained hardware such as edge devices running torch-CPU.

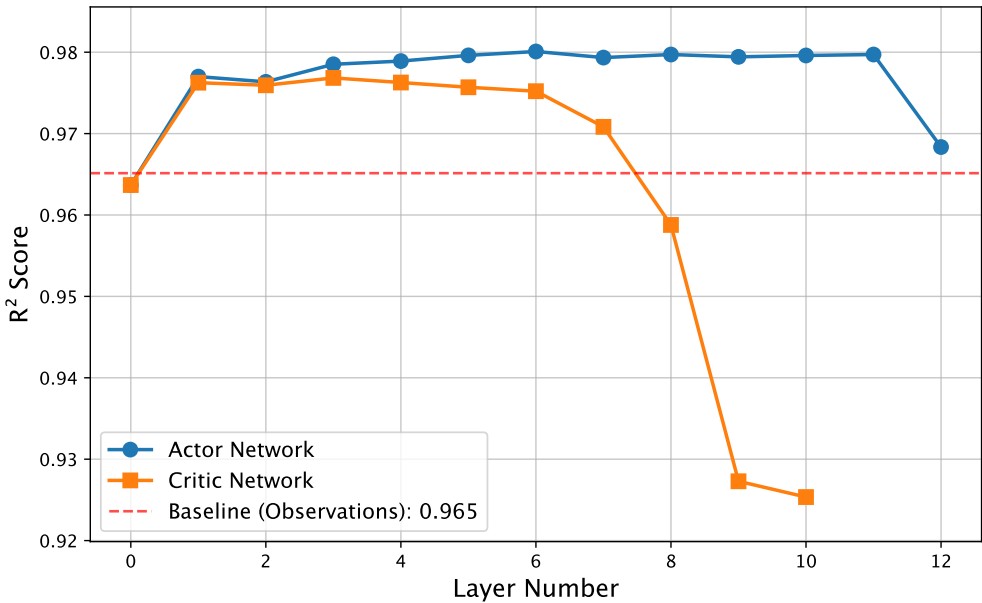

Figure 10: Action probing for the policy and value networks of a PPO agent trained on the inverted double pendulum environment. We report the $R^2$ scores between the predicted action and the model's action for relevant layers evaluated on a different set of activations. We compare with a baseline trained from the raw observations.

## 6.2  When to Use Other Frameworks

Despite this versatility, other interpretability frameworks can be a better fit in specific circumstances. `captum` remains unrivalled when an analyst requires an extensive catalogue of ready-made attribution algorithms and is not concerned with multi-input or multi-output settings. Transformer-centric libraries such as `transformer_lens` or `inseq` expose specialised utilities and visualisations that accelerate studies of language models at the expense of broader architectural support. Configuration-driven causal experimentation may be easier to conduct with `pyvene`, which provides configuration-based recipes for large-scale sweeps, while `zennit` offers broader implementations for LRP and its variants. Finally, `nnsight` bundles interpretability routines within a neat intervention API while enabling experiments to be run on models hosted on the NDIF servers.

## 6.3  Future Work

Future releases of TDHook will focus on three complementary directions. First, we will broaden the set of integrated methods by porting additional attribution, probing and causal-manipulation techniques, with an emphasis on more complex pipelines. This includes extending our work to more sophisticated models, like transformers with chain-of-thought modules, requiring large-scale analyses. Second, we plan to exploit the advanced features of `tensordict`, in particular memory-mapped tensors and zero-copy sharing, to further reduce the peak RAM footprint during large-scale analyses. Lastly, we aim to extend support to distributed and heterogeneous execution environments so that TDHook can seamlessly operate in multi-GPU and multi-node training loops. These technical improvements will be accompanied by richer documentation and end-to-end tutorials that showcase the library on vision, language and control tasks, with primary focus on composed interpretability pipelines.

**Broader Impact Statement**

As an interpretability tool, the framework proposed does not raise immediate ethical risks beyond those inherent to interpretability research. However, the goal of making interpretability methods easier to use and more accessible to non-experts can exacerbate misuse cases. The most concerning issues are:

- Creating misleading explanations if results are over-interpreted without rigour or without sufficient domain expertise.

- Model control abuse, e.g. using interpretability methods to bypass safety measures or to create harmful content.

Therefore, we encourage users to use the framework with caution and to be aware of the limitations of interpretability methods.

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

## A  Benchmarking Details

We present here more detailed results about the performance benchmarks presented in Section 4.

**Performance benchmark**   The performance benchmarking, summarised in Figure 4, consists of different tasks, each fitting its corresponding library, and all are available in the supplementary material (folder `scripts/bench/tasks`). We consider integrated gradients for `captum`, EpsilonPlus LRP rules for `zennit`, causal intervention for `nnsight` and activation caching for `transformer_lens`. Each task is run in isolation, meaning that a new process is created for each script, and no parallelisation is used. We launch our performance benchmark on an HPC cluster using SLURM with the following resources: 1 GPU A100 MIG 40GB, 7 CPU Intel Xeon Gold 6330 and 100GB of RAM. Relative performance details can be seen in Figure 11 and Figure 12 for the time metric, in Figure 13 and Figure 14 for the RAM metric, and in Figure 15 for the VRAM metric.

**Bundle size benchmark**   The bundle size benchmark, whose results are shown in Figure 5, measures the additional storage overhead of each framework. For each measurement, we install the specified package in isolation in a Python virtual environment managed by `uv` (Astral Software Inc, 2025). We then compute the memory and inodes occupied by the virtual environment and compare these measurements to having only `torch` installed.

## B  Use Cases Details

This appendix presents the models used in the use cases presented in Section 5, and other additional details.

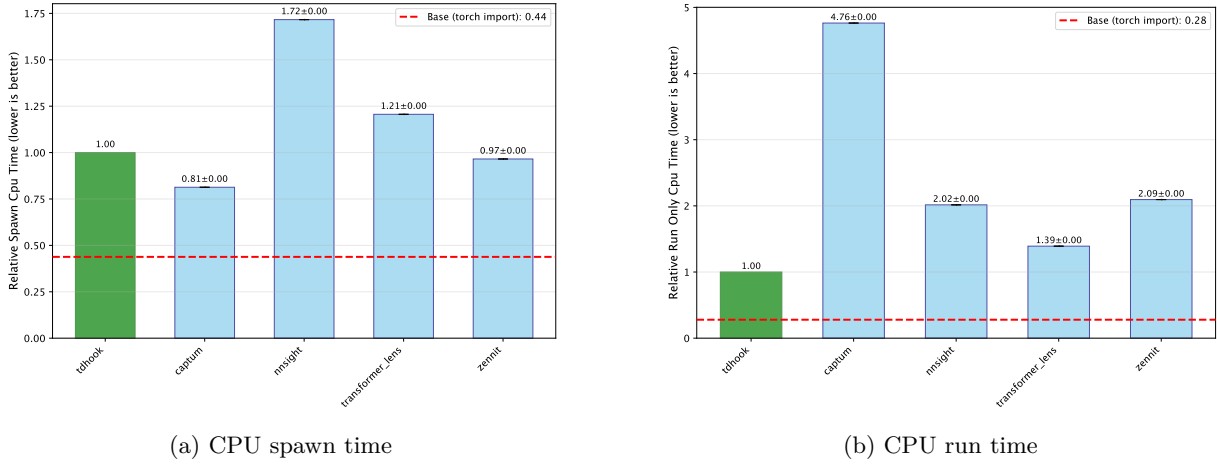

(a) CPU spawn time

(b) CPU run time

Figure 11: Relative time performance of CPU spawn and run across frameworks.

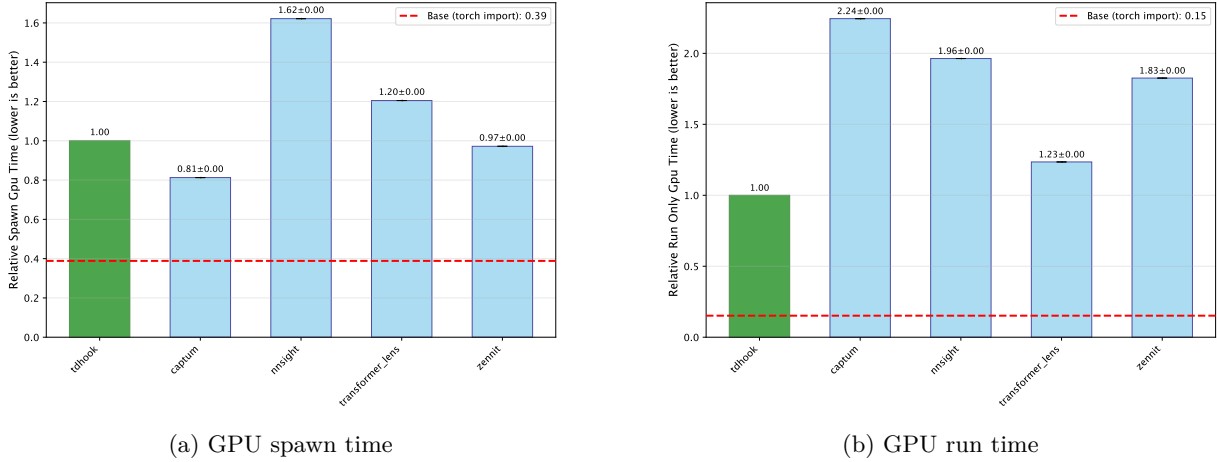

(a) GPU spawn time

(b) GPU run time

Figure 12: Relative time performance of GPU spawn and run across frameworks.

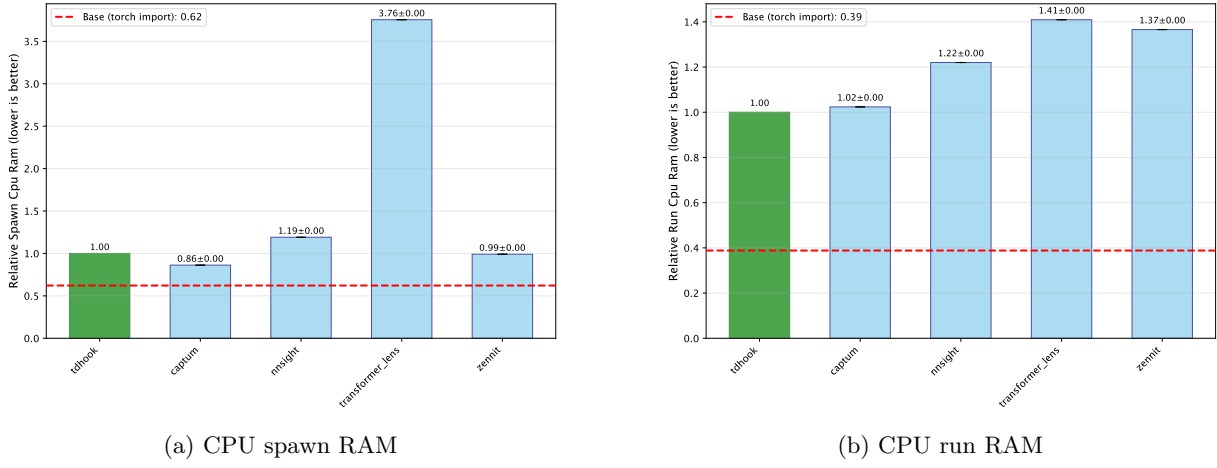

(a) CPU spawn RAM

(b) CPU run RAM

Figure 13: Relative RAM usage of CPU spawn and run across frameworks.

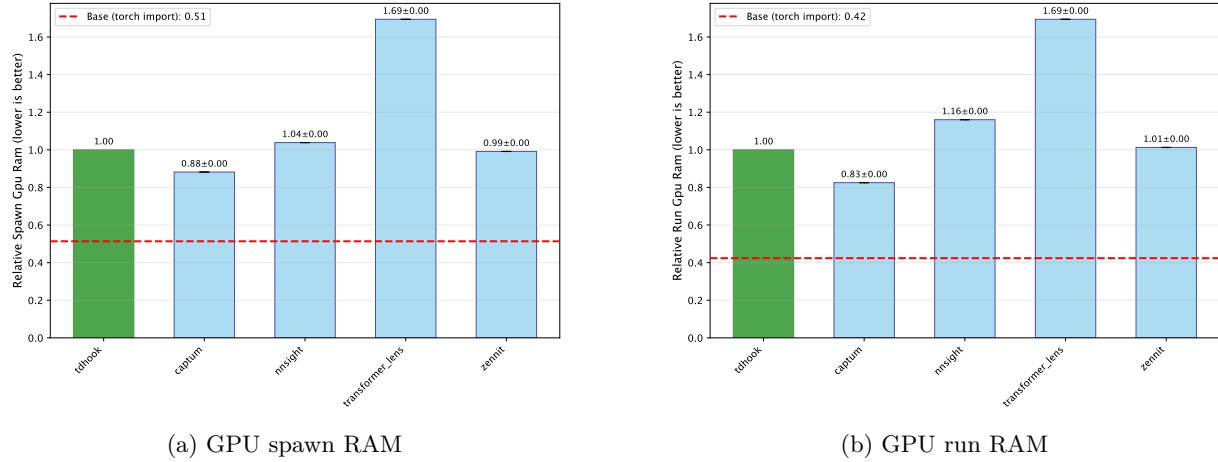

(a) GPU spawn RAM

(b) GPU run RAM

Figure 14: Relative RAM usage of GPU spawn and run across frameworks.

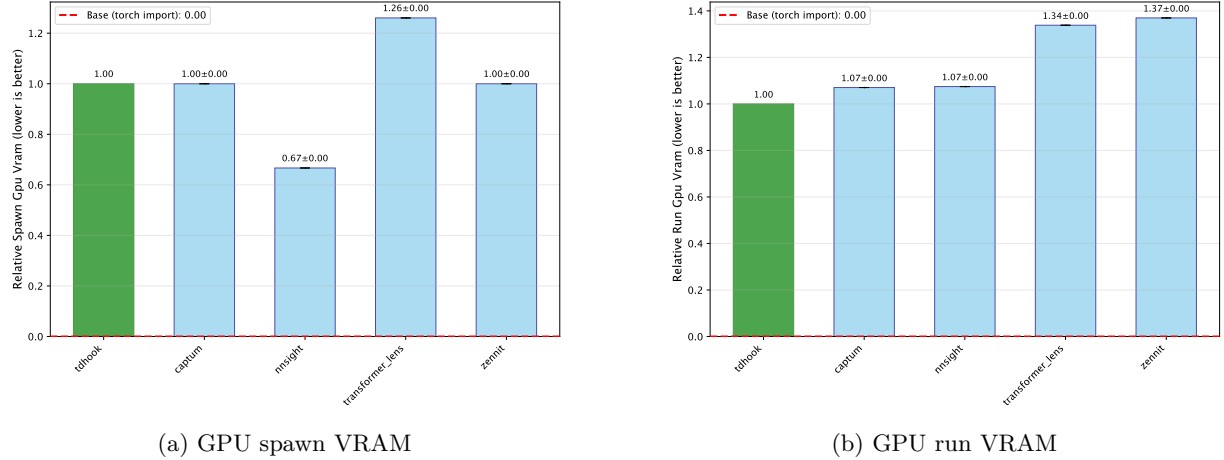

(a) GPU spawn VRAM

(b) GPU run VRAM

Figure 15: Relative VRAM usage of GPU spawn and run across frameworks.

**Concept attribution**    For our analysis, we choose the VGG16 model (Simonyan & Zisserman, 2014), loaded using the `timm` library (Wightman). We train linear probes for concepts on the texture dataset from (Cimpoi et al., 2013), using the `scikit-learn` library (Pedregosa et al., 2011). As pointed out in (Dreyer et al., 2023b), the difference of means (or signal CAV (Pahde et al., 2022)) is more effective to find latent concept directions, as shown in figures 16 and 17.

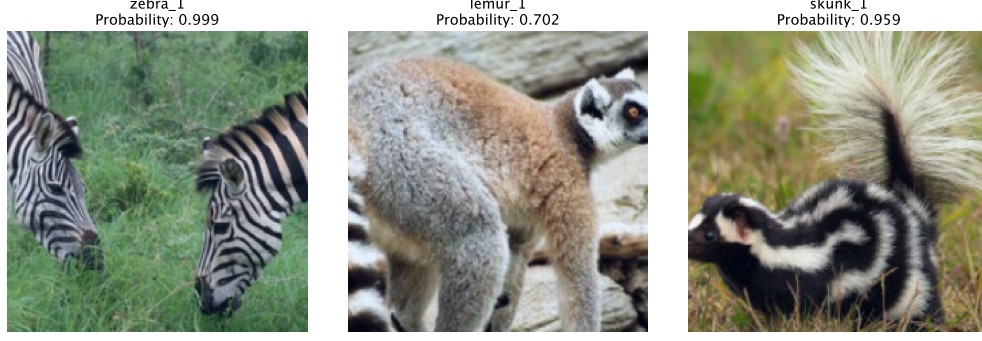

Figure 16: Logistic regression probe evaluation on the animal images.

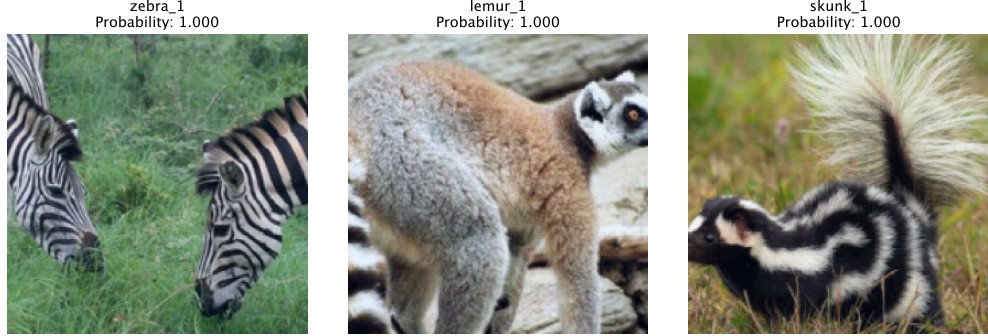

Figure 17: Mean difference probe evaluation on the animal images.

**Attribution patching**    We run our experiments using the GPT-2 model (Radford et al., 2019), loaded using the `transformers` library (Wolf et al., 2020). In order to compute Pearson correlation coefficients, we use the `scipy` library (Virtanen et al., 2020). We show the difference when using the AH-rule instead of the RelP method in figures 18 and 19.

**Multi-output model**    For studying a multi-output model, we choose a chess model trained with the AlphaZero algorithm (Silver et al., 2018). While the original proposed algorithm was closed-source, thanks to the Leela team, reproductions were made open source in the lc0 engine (The LCZero Authors). Further initiatives, like Maia networks (McIlroy-Young et al., 2020), include the development of human-like chess-playing agents. For our experiments, we choose the `maia-1900` network, which has two prediction heads, the policy head and the win-draw-lose head. These heads share a Squeeze-and-Excitation backbone (Hu et al., 2017), based on ResNet (He et al., 2015), made of 6 blocks.

**Composed model**    As we choose to train our agent with the PPO algorithm (Schulman et al., 2017), using the torchrl library (Bou et al., 2023), we require two networks: the policy and the value. These two networks are each a multi-layer perceptron with 6 hidden layers of 32 units each, activated by the hyperbolic tangent function. The model is trained on the inverted double pendulum environment (Todorov et al., 2012), from the gymnasium library (Towers et al., 2024), derived from cartpole (Barto et al., 1983). The agent is fully trained, with 1 million frames, before being studied.

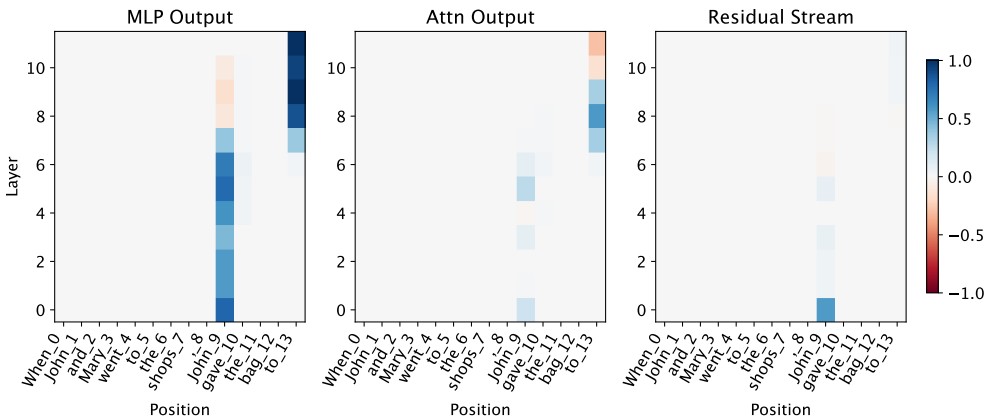

Figure 18: Attribution patching with the RelP method.

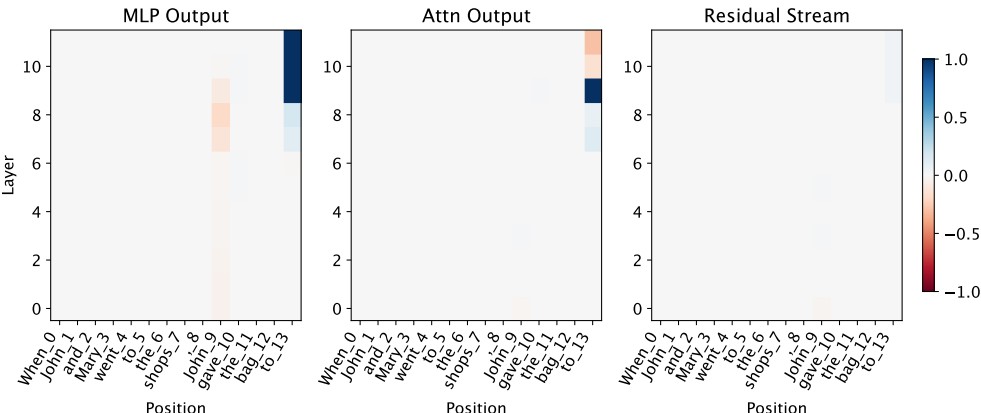

Figure 19: Attribution patching with the RelP method using the AH-rule.

# C   Simple Use Cases

This appendix showcases simple, self-contained examples of using TDHook for common interpretability tasks.

**Attribution**   TDHook provides a unified API for attribution methods, making it simple to switch between different techniques. In Listing 1, we demonstrate how to compute attribution maps for a pretrained VGG16 model. We highlight the minimal changes required to switch from `Saliency` (red lines) to `IntegratedGradients` (green lines), and present both results in Figure 20.

```python
import torch
import timm
from PIL import Image
from tensordict import TensorDict
from tdhook.attribution import Saliency
from tdhook.attribution import IntegratedGradients

# Load model and prepare image
model = timm.create_model("vgg16.tv_in1k", pretrained=True)
data_config = timm.data.resolve_model_data_config(model)
transforms = timm.data.create_transform(**data_config, is_training=False)

image = Image.open("results/simple/zebra_1.jpg").convert("RGB")
image_tensor = transforms(image)

# Define attribution target (zebra class = 340)
def init_attr_targets(targets, _):
    zebra_logit = targets["output"][..., 340]
    return TensorDict(out=zebra_logit, batch_size=targets.batch_size)

# Compute attribution
with Saliency(
with IntegratedGradients(
    init_attr_targets=init_attr_targets
).prepare(model) as hooked_model:
    td = TensorDict(
        {
            "input": image_tensor.unsqueeze(0),
            (" baseline", " input"): torch.zeros_like(image_tensor).unsqueeze(0)
        },
        batch_size=1,
    )
    td = hooked_model(td) # Access attribution with td.get(("attr", "input"))
```

Listing 1: Code example showing the switch from `Saliency` to `IntegratedGradients`.

**Steering Vectors**   TDHook supports intervention techniques such as steering vectors which have proven effective for language models (Rimsky et al., 2023). In Listing 2, we demonstrate how to extract a steering vector from GPT-2 that represents the concept of "wealth" and use it to steer the model's generation. When

completing the prompt "I work as a", the model is steered from "writer" to "pilot", reflecting the injected concept[4].

```python
from transformers import AutoTokenizer, AutoModelForCausalLM
from tensordict import TensorDict
from tdhook.latent import ActivationAddition, SteeringVectors

# Load model and tokenizer
model = AutoModelForCausalLM.from_pretrained("gpt2")
tokenizer = AutoTokenizer.from_pretrained("gpt2")

# Prepare inputs
positive_inputs = tokenizer.encode("I am rich.", return_tensors="pt")
negative_inputs = tokenizer.encode("I am poor.", return_tensors="pt")
base_inputs = tokenizer.encode("I work as a", return_tensors="pt")

# Extract steering vector (rich - poor)
with ActivationAddition(["transformer.h.7.mlp"]).prepare(model) as hooked_model:
    td = TensorDict(
        {
            ("positive", "input"): positive_inputs,
            ("negative", "input"): negative_inputs
        },
        batch_size=1
    )
    td = hooked_model(td)

steering_vector = td.get(("steer", "transformer.h.7.mlp")).sum(dim=0)

# Define steering function
def steer_fn(module_key, output):
    return output + 4 * steering_vector

# Apply steering during inference
with SteeringVectors(["transformer.h.7.mlp"], steer_fn=steer_fn).prepare(model) as hooked_model:
    td = TensorDict({"input": base_inputs}, batch_size=1)
    td = hooked_model(td)

# Compare results
steered_token = td.get(("output", "logits")).max(dim=-1).indices[0, -1]
original_token = model(base_inputs)["logits"].max(dim=-1).indices[0, -1]

print(f"Steered: {tokenizer.decode(steered_token)}")  # Output: "pilot"
print(f"Original: {tokenizer.decode(original_token)}")  # Output: "writer"
```

Listing 2: Code example showing the extraction and application of a steering vector in GPT-2.

---

[4]Additional samples could be used to extract more robust and representative steering vectors.

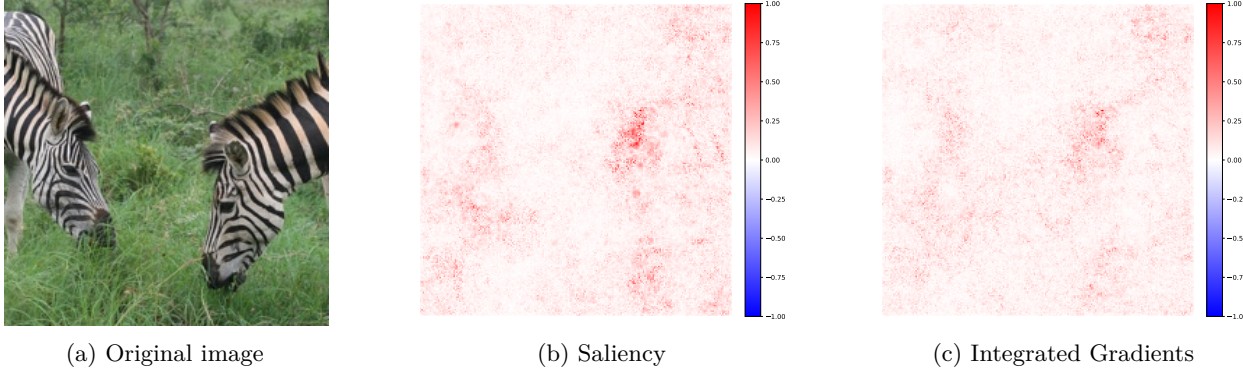

(a) Original image          (b) Saliency          (c) Integrated Gradients

Figure 20: Comparison of attribution maps for the "zebra" class (index 340) using VGG16. Saliency (left) and Integrated Gradients (right) are computed using the code above.

## D Codebase Outline

This appendix provides an overview of the codebase distributed with the supplementary material.

**Source code** The source code resides in the `src/tdhook` directory, while the test suite mirrors the same structure under `tests`. The package is organised around three core modules:

- `module.py` defines the `HookedModel` wrapper among others, which is the core class that wraps a PyTorch model and provides the get-set API or the run context via the `HookedModuleRun` class.

- `hooks.py` gathers utility functions to create and manipulate hooks, as well as proxy classes to provide cache flexibility.

- `contexts.py` groups context managers that serve as building blocks for the different methods; they enable the creation and management of hooks and modules.

Higher-level methods are grouped in dedicated sub-packages: `attribution` for attribution methods, `latent` for latent manipulation methods, and `weights` for weights-based methods.

**Main classes interactions** Each ready-to-use method inherits from the `HookingContextFactory` class, which is stateless, and is responsible for preparing the initial model by rewriting its forward pass when necessary, composing it with other `TensorDictModule` instances and registering the necessary hooks. The `HookingContextFactory.prepare` method creates a `HookingContext` instance that spawns a bound `HookedModel` inside the context and ensures its proper restoration after the context is exited. The `HookedModel` class provides a powerful API on its own to manage hooks and cache directly, and is thus a great point of entry to customise any method. The `HookedModel.run` method creates a `HookedModuleRun` instance, which enables the definition of transparent intervention schemes with automatic module execution, hook (de)registration and cache management.

**Ready-to-use methods** Table 2 provides a summary of all implemented methods organised by category.

Attribution methods are implemented in the `attribution` package:

- `gradient_attribution/saliency.py`: gradient attribution (Simonyan et al., 2013) and its gradient-times-input variation (Shrikumar et al., 2016).

- `gradient_attribution/integrated_gradients.py`: integrated gradients (Sundararajan et al., 2017) and its conditional variant (Dhamdhere et al., 2018).

- `gradient_attribution/grad_cam.py`: grad-CAM (Selvaraju et al., 2016).

Table 2: Summary of implemented methods by category

| Category | Count | Methods |
| --- | --- | --- |
| Attribution | 13 | Simonyan et al. (2013) Shrikumar et al. (2016) Sundararajan et al. (2017) Dhamdhere et al. (2018) Selvaraju et al. (2016) Bach et al. (2015) Lapuschkin et al. (2019) Montavon et al. (2019) Andéol et al. (2021) Montavon et al. (2015) Achtibat et al. (2022) Mahendran & Vedaldi (2015) Springenberg et al. (2014) |
| Latent Manipulation | 8 | Chen et al. (2020) Abnar & Zuidema (2020) Alain & Bengio (2018) Kim et al. (2018) Vig et al. (2020) Belrose et al. (2023) Dreyer et al. (2023b) Rimsky et al. (2023) |
| Weights | 6 | Meng et al. (2022) Cunningham et al. (2023) Dunefsky et al. (2024) Ilharco et al. (2022) Yeom et al. (2019) Pochinkov & Schoots (2024) |

- `gradient_attribution/lrp`: different LRP rules such as LRP-0, LRP-$\epsilon$ z-plus (Bach et al., 2015), flat (Lapuschkin et al., 2019), gamma (Montavon et al., 2019; Andéol et al., 2021), w-square (Montavon et al., 2015) and its conditional variant (Achtibat et al., 2022).

- `activation_maximisation.py`: activation maximisation (Mahendran & Vedaldi, 2015).

- `guided_backpropagation.py`: guided backpropagation (Springenberg et al., 2014).

Latent manipulation methods are implemented in the `latent` package:

- `activation_caching.py`: maximally activating samples (Chen et al., 2020) and attention visualisation (Abnar & Zuidema, 2020).

- `probing.py`: linear probing (Alain & Bengio, 2018) and concept activation vectors (Kim et al., 2018).

- `activation_patching.py`: causal mediation analysis (Vig et al., 2020) and latent editing (Belrose et al., 2023; Dreyer et al., 2023b).

- `steering_vectors.py`: steering vectors (Rimsky et al., 2023).

Weights-related methods are implemented in the `weights` package:

- `adapters.py`: ROME (Meng et al., 2022), sparse autoencoders (Cunningham et al., 2023) and transcoders (Dunefsky et al., 2024).

- `task_vectors.py`: task vectors (Ilharco et al., 2022).

- `pruning.py`: relevance-based pruning (Yeom et al., 2019) and circuit pruning (Pochinkov & Schoots, 2024).

Future work could extend attribution with occlusion-based methods (Zeiler & Fergus, 2013), latent activation with AtP* (Kram'ar et al., 2024), or weights-based methods with crosscoders that generalise adapters to multiple layers and models (Lindsey et al., 2024).

**Scripts** Executable scripts are located in the `scripts` directory, and their outputs, like figures or logs, are stored in the `results` directory. The `scripts/bench` subdirectory hosts the benchmarking pipeline, with scripts to run the different methods and plot the results. In particular, each task has corresponding scripts, one per library and per task, in the `scripts/bench/tasks` directory, which define variations of the task, such as the batch size or model hyperparameters. The bundle size benchmark utilises an empty project `scripts/bundle_test` to install each package in isolation. The example use-cases used in Section 5 are located in the `scripts/torchrl` and `scripts/lczerolens`, respectively.

# E  Software

This appendix summarises the software used in this work.

**Library core**   First, in order to build our library, we heavily relied on `torch`, (Ansel et al., 2024) as a building block for manipulating models and tensors, at the heart of our library are the `torch` hooks. And as previously mentioned, we focused our efforts around the `tensordict` library (Bou et al., 2023), using `TensorDict` to manipulate the different artifacts and `TensorDictModule` to implement the different methods.

**Library tests**   We ran all our tests using `pytest` (Krekel et al., 2004). In addition, we used some of the aforementioned libraries to compare our implementation for different methods. LRP rules were compared against `zennit` implementations (Anders et al., 2021). Attribution methods were compared against `captum` implementations (Kokhlikyan et al., 2020) and the get-set API was compared against `nnsight` (Fiotto-Kaufman et al., 2024) using different intervention schemes.

**Scripts**   For demonstrating use cases of the library, we used `timm` (Wightman) to load vision models, `transformers` (Wolf et al., 2020) to load language models, `torchrl` (Bou et al., 2023) to train a simple RL agent using the PPO algorithm and `lczerolens` (Yoann Poupart, 2024) to manipulate the chess networks. The linear probes were trained using the linear regression model from the `scikit-learn` library (Pedregosa et al., 2011). In order to compute Pearson correlation coefficients, we used the `scipy` library (Virtanen et al., 2020). For plotting our benchmarking and study results, we used `matplotlib` (Hunter, 2007) and `seaborn` (Waskom, 2021). For manipulating the experiment results, we used `pandas` (pandas development team, 2025) and `numpy` (Harris et al., 2020). We used `uv` (Astral Software Inc, 2025) to run our scripts and manage our dependencies.

