# OpenReview forum: "TDHook: A Lightweight Framework for Interpretability"
_TMLR — Rejected by TMLR_

### Review · Reviewer_z6te · 2025-11-23

**Summary Of Contributions:**

Strengths:

- Engineering contribution: The paper introduces a new interpretability framework, TDHook, which provides methods for attribution, latent manipulation, and weight manipulation, and is ready to use across different domains (e.g., CV, RL).

- The authors compare TDHook with several existing interpretability frameworks and present various use cases for TDHook.

- The writing is solid, and most statements are well justified.

Weaknesses:

- A disappointment is the lack of demos involving transformer attention, generation, chain-of-thought modules, and similar components for language modeling. Since there is clear overlap between TDHook and existing libraries for model explanation, the claim of being “generic” is overstated if the framework is not evaluated on widely used large transformer architectures.

- For large models such as LLMs, issues related to memory (e.g., storing activations for long contexts), large parameter footprints, and structured generation outputs are critical but remain unaddressed (in the experiments).

- The contribution appears to lie primarily in tooling rather than in advancing interpretability algorithms. Although TMLR values soundness over subjective notions of novelty, the reviewer does not find the submission to meet the standard for a scientific contribution appropriate for TMLR.

- The reported improvements in bundle size are useful, but interpretability frameworks are often deployed in settings where memory overhead is less of a concern than correctness or reliability. It remains unclear whether TDHook provides more consistent or more reliable results compared to existing frameworks. Would `tdhook` provides consistent or more reliable results than other frameworks?

**Additional Comments:**

N/A

**Audience:**

No

**Audience Explanation:**

The tooling or package can be found on GitHub and should be easy to locate in the documentation. As a reviewer, I do not see a clear need for this submission within the TMLR audience.

**Broader Impact Concerns:**

The authors have already addressed sufficiently (after Sec 6.3) in the paper.

**Claims And Evidence:**

Yes

**Claims Explanation:**

Although I selected “Yes”, not all of the stated contributions are fully supported.

**Requested Changes:**

Just a few minor suggestions for writing.

- In the Attribution Methods paragraph (Background), the statement “The key idea (of attribution methods) is to compute the
sensitivity of the output or any intermediate representation with respect to the input or other intermediate
representation.” could be revised. Not all attribution methods rely on gradients or sensitivity; for example, perturbation-based approaches operate without gradient information.

- In Section 2.1, the definition of weight-based methods is vague. The authors could improve this section by providing a clearer definition or by explicitly referencing the relevant line of work.

---

> ### Comment · Action_Editor_EMVy · 2025-11-24
> **Clarifications needed**
>
> Dear Reviewer z6te,
>
> Thank you for submitting the review.
>
> Could you please clarify what claims made in the submission are (or NOT) supported by accurate, convincing and clear evidence?
>
> Thank you,
> Your AE

---

> > ### Comment · Reviewer_z6te · 2025-11-24
> >
> > Dear AC,
> >
> > Only the claim of "generic framework" is not supported (as pointed out in Weaknesses #1).

---

> > > ### Comment · Action_Editor_EMVy · 2025-11-24
> > >
> > > Dear Reviewer z6te,
> > >
> > > Thank you for the quick response.
> > >
> > > Could you also clarify what claims are supported by what evidence?
> > >
> > > Best,
> > > Your AE

---

> ### Author Response · Authors · 2025-11-26
>
> We thank the reviewer for the thoughtful feedback. We address each concern below.
>
> **Transformer/LLM demonstrations and “generic” claim**
> TDHook is model-agnostic and supports any PyTorch module, including transformers. In the current submission, we applied TDHook to the GPT-2 suite for benchmarking against `transformer_lens` and `nnsight`. These experiments are briefly mentioned in Appendix A but not highlighted in the main text. We will clarify this in the revised version and expand the description of how TDHook handles transformer architectures. While we did not include chain-of-thought modules, we agree that such demonstrations would strengthen the paper and fall within TDHook’s scope. Thank you for pointing this out.
>
> **Memory and activation storage for large models**
> We benchmarked resource usage (CPU/RAM/GPU) of different model sizes (up to 1B parameters) with different batch sizes (up to 100k for MLPs), including different transformer models (GPT-2 suite). We will add explicit discussion of these results, highlighting TDHook’s mechanisms for memory-efficient activation caching, which enable it to handle larger models and batches.
>
> **Scientific contribution vs. tooling**
> We acknowledge that TDHook is a practical toolkit. However, its scientific contribution lies in three aspects: (i) a unified abstraction for attribution, latent, and weight interventions, (ii) empirical analyses highlighting failure modes in existing libraries, and (iii) extended analyses of complex use cases.
>
> **Reliability and consistency**
> TDHook does not make methods “more reliable” than existing frameworks, as reliability primarily depends on the method itself, but we ensure consistency by quantitatively testing against existing implementations. However, by spanning multiple categories of interpretability (attribution, latent, weight), TDHook ease the development of complex use cases that can be hard to implement in other frameworks.
>
> **Minor writing clarifications**
>
> - **Attribution methods:** We maintain the use of “sensitivity” as used by Zeiler & Fergus (2013). Occlusion-based methods measure output variation with respect to input variation without requiring a total derivative.
> - **Weight-based methods:** Thank you for this suggestion. We will clarify the definition and reference additional literature, emphasising interventions on parameters.
>
> **Scope of revisions**
> We will clarify and expand the discussion of TDHook’s transformer experiments, memory benchmarking, and improve definitions in the main text to address visibility and conceptual framing.

---

### Review · Reviewer_puvt · 2025-11-24

**Summary Of Contributions:**

Proposed a interpretability software based on just two main libraries. The code is accessible with 20+ use cases like multi-input and output. Provides nice visuals to show inner workings and compare with existing libraries.

**Audience:**

Yes

**Audience Explanation:**

Absolutely, because the proposal is generic method which every researcher is interested in.

**Broader Impact Concerns:**

It would be nice if you could comment on a non-torch library use case as well.

**Claims And Evidence:**

Yes

**Claims Explanation:**

The breath of use cases discussed and benchmarking is evidence for the claims made. The cited references are sufficient. Prior methods and libraries discussed is noteworthy effort.
The claim that the software works for 25+ use case is listed in detail in Appendix.
The claim about comparison with other softwares is supported by benchmarking section - establishing the lightweight claim.
The claim that multi-step interpretability probing like causal mediation analysis is also implemented in the software.

**Requested Changes:**

1. it is not clear what coverage means in Table 1
2. Benchmarking is missing benchmarking by model size. or does it even matter?
3. In Section 5.1, concept attribution, it is not clear if the "concept" is like a conditional extra input provided to the interpretation software.
4. Figure 8 should have error bars.

---

> ### Comment · Action_Editor_EMVy · 2025-11-24
> **Clarifications needed**
>
> Dear Reviewer puvt,
>
> Thank you for submitting the review.
>
> Could you please clarify what claims made in the submission are (or NOT) supported by accurate, convincing and clear evidence?
>
> Thank you,
> Your AE

---

> ### Author Response · Authors · 2025-11-26
>
> We thank the reviewer for the positive feedback and constructive suggestions. We address each point below.
>
> **Table 1 “coverage”**
> Coverage in Table 1 refers to the percentage of code that has been tested. We will clarify this definition explicitly in the table caption and main text to avoid ambiguity.
>
> **Benchmarking by model size**
> Our performance benchmarks already include comparisons across different model sizes (up to 1B parameters). For MLP experiments, we vary both width and depth, and for transformer experiments, we use multiple sizes from the GPT-2 family. We agree this could be clearer in the main text.
>
> **Concept attribution (Section 5.1)**
> The term “concept” that we use for our experiment is user-defined and often refers to a direction in the latent space of a model layer. Concepts are generally computed post-hoc using CAVs, probes, or individual/groups of neurons, but can also involve more complex transformations, such as sparse auto-encoders. We will clarify this in the revised text to make it explicit.
>
> **Figure 8 error bars**
> We can extend our experiments to account for performance variability across different settings and improve reliability.
>
> **Non-PyTorch use case**
> TDHook currently supports PyTorch only. Extending to other frameworks like TensorFlow or JAX would require duplicated implementations due to framework-specific differences, similar to approaches used by the `transformers` library. We acknowledge this limitation and note that supporting multiple frameworks is an interesting direction for future work, especially with our high-level user interface. Thank you for pointing this out.
>
> **Scope of revisions**
> We will clarify definitions (coverage, concepts), highlight model-size benchmarking, and add error bars to improve readability and precision.

---

### Review · Reviewer_91yW · 2025-11-24

**Summary Of Contributions:**

This paper describes TDHook, a library that implements a suite of  25 neural network interpretability methods from the literature integrated with PyTorch's `tensordict` abstraction. TDHook can be easily applied to general PyTorch `module`s in just a few lines of code. The paper compares TDHook with existing libraries descriptively, in terms of their features, and via performance benchmarks (which also serve to test its correctness compared to existing implementations of the methods). The paper showcases use cases of TDHook with composite pipelines and models.

**Audience:**

No

**Audience Explanation:**

The paper doesn't contain "findings" in the strict sense, since I don't see the analyses as providing new insights on the methods that tdhook implements.

To be clear, this is not to say that the library cannot be useful for some individuals in the TMLR audience. But the criteria for a library being useful are different, and perhaps not captured by a regular scientific paper peer review (users will judge better how usable the library is, how easily they can adapt it to their use cases, etc). By reading the paper itself, I'm unsure if TMLR readers will gain insights on the topic.

**Claims And Evidence:**

No

**Claims Explanation:**

The paper focuses on describing the new library, `tdhook`, and not on new analyses or methods. This is a non-traditional submission to a place like TMLR - I could not find a similar tool paper in TMLR to take as a reference, so frankly I'm unclear about the standards to judge such a paper. However, taking the stated contributions at face value, some are still hard to verify:

* The library submitted with the artifact has essentially no documentation. Appendices C and D of the paper describe the code base at a high-level. But for the library to be useful, and for me to verify that one can use it "only requiring a few lines of code to set up", it would rather have concrete usage examples such as taking an existing vision/language model and running simple analyses on them with little setup, as advertised.
* For the "Concrete use cases of the TDHook", I found the description of the use cases still quite vague:
  * For the first use case (with the racoon), I do not know (a) what model is being analyzed, (b) trained for what task (I'm assuming image classification), (c) where does the concept "striped" come from (is it in natural language? or just post-hoc labeling of a concept activation vector?), (d) how do I conclude from Figure 7 that TCAV is "more faithful", and (e) what was the qualitative analysis suggesting "attribution is not localised" (typo: localized)?
  * The second use case describes attribution patching with two conflicting definitions. Then, I don't know what exactly is being analyzed.
  * The use cases with composite models are a bit more clear, at least for chess. In "Analyze composed models" (should be analyze), there is a typo: "an each agent can have" -> "and each agent can have". Moreover, the conclusion of this part is unclear to me: "For the policy, the last layers contain less information about the predicted action because of the compression and randomness of actions, since the action is sampled from a distribution. For the value, the information about the action is not trackable in the last layers, and the probe overfitted on the training data.". I don't understand the intuition for why the policy layers lose information about the action (isn't the action predicted from the last layer of the policy?). Also, is the overfitting of the probe a "user error" here or something happening internally in tdhook that is undesirable? How would users know?

**Requested Changes:**

At least the concrete use cases should be drastically clarified in the main paper (see above). It would also be helpful to have the complete implementation of a few simpler examples in the paper itself (even if in the appendix), to help readers have a concrete sense of how the library is used.

---

> ### Author Response · Authors · 2025-11-26
>
> We thank the reviewer for the detailed feedback. We address each concern below.
>
> **Library usage examples/documentation**
> We acknowledge that the current artifact’s documentation is missing, but we would like to mention that Figures 2 and 3 present example usage. We can add more small, self-contained examples demonstrating how TDHook can be applied to vision and language models in just a few lines of code in the appendix.
>
> **First use case (raccoon / TCAV example)**
> The model analysed is VGG16, trained on the ILSVRC object detection dataset, while this is described in Appendix B, it should be referenced in the main paper. Concepts like “striped” are defined post-hoc, using CAVs, activation maximisation, and relevance maximisation applied to the same set of images from the texture dataset from Cimpoi et al. (2013). CAV marginally outperforms the other methods according to pixel-flipping evaluation, as shown in Figure 7. Yet, qualitatively, high saliency is sometimes less localised to the concept (e.g., the lemur’s tail), which we will clarify in the text.
>
> **Second use case (activation/attribution patching)**
> We apologise for the confusion. The first term “Attribution Patching” should be replaced by Activation Patching, the original method. Attribution Patching is an approximation using a first-order Taylor expansion. For this experiment, our contribution lies in extending recent work showing that replacing gradients with saliency methods (e.g., LRP) can improve accuracy. We will clarify the definitions and experimental setup in the main text.
>
> **Composite models/chess example**
>
> - We believe that "Analyse" is correct, given that the paper is written using British English.
> - The last policy layer loses information because it is highly compressed (predicting only the mean and std of a Tanh-Normal distribution). This is inherent to the model architecture, not a limitation of TDHook.
> - Probe overfitting is expected: training a linear probe on latent activations can overfit naturally due to the limited semantic information present. Therefore, the final test accuracy reflects the amount of relevant information in the latent space rather than a model accuracy; thus, it is not a user error or TDHook issue.
>
> Thank you for engaging with the results.
>
> **Scope of revisions**
> We will fix typos, add concrete usage examples and clarify concepts, methods, and evaluation metrics for the raccoon and patching use cases. We will also explain the intuitions behind information loss and probe behaviour in composite models.

---

> > ### Comment · Reviewer_91yW · 2025-12-01
> >
> > Thank you for the response. I took a second read at the use cases informed by your response, and I think I understand the Concept Attribution experiment now, as well as the point of the second experiment. Integrating your clarifications into the text will certainly be very helpful. Indeed these experiments, if well described, make clear the point that you make it very easy to swap one method for another and measure the impact of the change.
> >
> > I look forward to looking at the revision of the paper once the authors have it - especially other simple examples of the library and the description of the case studies.

---

> > > ### Author Response · Authors · 2025-12-02
> > >
> > > We have submitted our revised manuscript. We would be happy to address any remaining questions or concerns or to hear any additional feedback you might have.

---

### Comment · Action_Editor_EMVy · 2025-11-25
**Start author-reviewer discussions**

Dear authors,

You have three reviews. Please address their comments as soon as possible. The discussion period will end on December 7th, 2025.

---

Dear reviewers,

Thank you for submitting your reviews. As this paper received three reviews, now it's time to have discussions with authors.

You will be able to submit your formal decision recommendation starting in 2 weeks. Your prompt responses to authors' comments would be very appreciated.

Remember that different from other journals / conferences, [TMLR's acceptance criteria](https://jmlr.org/tmlr/reviewer-guide.html) are based on positive answers to the following two questions

- Are the claims made in the submission supported by accurate, convincing and clear evidence?
- Would some individuals in TMLR's audience be interested in the findings of this paper?

Some of you mentioned novelty of the method, but novelty of the studied method is not a necessary criteria for acceptance at TMLR.

It's still not too late to review [the TMLR's reviewer guideline](https://jmlr.org/tmlr/reviewer-guide.html) in case you are not aware of that.


Thank you!
Your AE

---

### Author Response · Authors · 2025-12-02
**New revision submitted**

We thank the reviewers for their thoughtful and constructive feedback, which has helped improve the clarity, structure, and accessibility of the paper.

In the revised version, we clarified the presentation of our transformer experiments, expanded the discussion of memory and performance benchmarking, and improved explanations of the concept attribution and attribution patching use cases. We also added two concise examples in the appendix: a simple attribution workflow on VGG16, and a minimal steering-vector intervention on GPT-2, to illustrate how TDHook can be applied in practice.

We welcome any additional feedback and would be happy to provide further clarifications if needed.

---

### Comment · Action_Editor_EMVy · 2025-12-09
**Submit your official recommendation**

Dear Reviewers,

Now it's time for you to submit an official recommendation for this submission.

The authors responded to your initial reviews.

Please
1. consider their rebuttal comments,
2. respond to the comments ***ASAP*** if you have follow-up questions or comments, and
3. submit your official recommendation ***by December 22nd***, including ***your thoughts on the author rebuttal***, specifically, what point in your review their responses addressed AND did not address. It will greatly help me write a meta-review and make a fair decision.

Thank you,
Your AE

---

### Decision · Action_Editor_EMVy · 2026-01-04

**Recommendation:** Reject

**Additional Comments:**

This is not a basis for rejection, but just a comment that a property that TDHook has minimal dependencies is not a significant contribution.
There already exists [a package](https://github.com/yoshitomo-matsubara/torchdistill/) built on PyTorch and supports torch-based forward hook manager to seamlessly extract input/output of an arbitrary torch module
- [torchdistill: A Modular, Configuration-Driven Framework for Knowledge Distillation](https://arxiv.org/abs/2011.12913)
- [torchdistill Meets Hugging Face Libraries for Reproducible, Coding-Free Deep Learning Studies: A Case Study on NLP](https://aclanthology.org/2023.nlposs-1.18/)

**Audience:**

No

**Audience Explanation:**

The paper is focused on a tool and framework designs. Two reviewers find that TMLR audience would not be interested in this work.

**Claims And Evidence:**

No

**Claims Explanation:**

The authors' response to Reviewer z6te's comment claims its scientific contributions, but those were not testable claims.
For use cases, it looks like just showing outputs of their reimplemented methods rather than validating their implementations by comparing to the same methods available in existing frameworks

We found the following claim testable
> TDHook makes it easier to compose complex interpretability pipelines across diverse domains, as demonstrated by our use cases study

But, we did not find any accurate, convincing and clear evidence to support the claim.
The use cases that the author demonstrated are presented in Section 5, which I find not conclusive or supporting the claim either